# Additive Based on Synthetic Aluminosilicates for Dry Lime Construction Mixtures

**Valentina Ivanovna Loganina ***, **Kristina Vladimirovna Zhegera and Maria Anatolyevna Svetalkina**

Department "Quality Management and Technology of Construction Production", Penza State University of Architecture and Construction, st.G. Titova, 28, 440028 Penza, Russia
* Correspondence: loganin@mail.ru; Tel.: +7-890-9316-9950

**Abstract:** The possibility of increasing the durability of coatings based on lime dry construction mix by introducing an additive containing synthetic aluminosilicates is substantiated. The regularities of the structure formation of the lime composite in the presence of an additive containing synthetic aluminosilicates, which additionally consists of a formation of calcium–sodium hydrosilicates and minerals of the zeolite group, an increase in the amount of chemically bound lime by 8.74%, are revealed. X-ray diffraction analysis and thermodynamic calculations have established that the mineralogical composition of the crystalline phase of the additive based on synthetic aluminosilicates is represented by thenardite, gibbsite, and the minerals of the zeolite group. It is shown that the content of the amorphous phase is 77.5%. It was found that the additive based on synthetic aluminosilicates is characterized by high activity, which is more than 350 mg/g. It was also found that the introduction of an additive based on synthetic aluminosilicates into the formulation of a lime dry mixture accelerates the curing of coatings and increases the compressive strength after 28 days of air-dry hardening by 1.9 times.

**Keywords:** additives; synthetic aluminosilicates; synthesis; properties; lime mix

## 1. Introduction

The modern production of dry building mixtures is unthinkable without the use of modifying additives. Even though the main processes for the formation of the properties of mortars are determined by interactions in the "mineral binder—aggregate—water" system, the introduction of inorganic and organic modifying additives into such a system makes it possible to change almost all characteristics of the material.

As is known, lime binder hardens very slowly, which complicates the production of finishing works. To accelerate the hardening and increase the strength of lime composites, additives are introduced into the formulation. Zeolites are one of the common additives. Considering the locality of natural zeolite reserves and the heterogeneity of their composition and properties, the use of synthetic zeolites in lime compositions is proposed.

Zeolites are aqueous aluminosilicates of alkali and alkaline earth metals with an open-frame cavity structure. The crystalline three-dimensional framework consists of aluminum–silicon–oxygen [(Si,Al)O$_4$] tetrahedra combined into simple, double, and more complex rings; moreover, each ring includes 4, 5, 6, 8, or more tetrahedra. Because some of the tetravalent silicon ions (Si$^{+4}$) are replaced by trivalent aluminum ions (Al$^{+3}$), this frame has a negative charge, compensated by the presence on the walls of the cavities of one- and two-tape cations of sodium, potassium, calcium, and magnesium, as well as, less often, barium, strontium, lithium, and other metals.

Total volume of cavities and channels connecting them in zeolites constitutes about 50% of the crystal volume, and the diameter of these channels' crystal surface varies from 0.26 to 0.8 nm. Internal cavities and the channels connecting them are filled with molecules of the so-called "zeolitenoah" of water.

In general, the composition of zeolites can be expressed by the formula:

$$M_x D_y [Al_x + 2y Si_z O_2 x + 4y + 2z] \, nH_2O,$$

where M and D are mono- and divalent cations, respectively;

x + y is the sum of tetrahedra in a unit cell.

Synthetic zeolites can be obtained using the hydrothermal crystallization of alkaline aluminosilicate mixtures. There are several ways to obtain synthetic zeolites. In 1862, St. Clair Deville received a synthetic potassium phillipsite when heating silicate and potassium aluminate at 200 °C. Synthetic chabazite was obtained under similar conditions at temperatures up to 170 °C. In 1948, Professor R. Barrer began work on the synthesis of zeolites [1]. Breck D. et al. determined the conditions for obtaining basic zeolites general purpose type NaX and NaA [2,3]. One way to obtain zeolites is through the continuous hydrothermal crystallization of aluminosilicate [4]. There is also a method for obtaining zeolites of the NaX type [5]. This way consists of preparing solutions of sodium metasilicate and sodium aluminate, adding an amine-containing compound from the series triethanolamine, polyethylene polyamine, or phenylenediamine to solutions, and preparing a gel by mixing the resulting solutions. The gel is mixed with dimethyl sulfoxide, and hydrothermal crystallization is carried out at a temperature of 70–100 °C. In [6,7], the technology for preparing adsorbents includes the preparation of hydro reactive heterogeneous composition containing aluminum powder, sodium metasilicate, and water, and the interaction of the initial components. Another method for producing sodium aluminosilicate is known, which involves the interaction of a solution of sodium silicate and an aluminum salt in a mixture of water-immiscible tributyl phosphate [8,9].

In Russian patent 2476378 No., the NaY zeolite was prepared using an aqueous mixture of sodium oxide, silica, and alumina, containing 1–5.0 M phenylenediamine, then heating the mixture to a crystallization temperature, maintaining this temperature until the crystals and the separation of the crystals from the solution were achieved. The mole ratio of oxides in the mixture were $Na_2O/SiO_2$ 1.6–2.2; $SiO_2/Al_2O_3$ 1.5–5; $H_2O/Na_2O$ 70–150. The invention provides a faujasite structure zeolite—NaY with a ratio $SiO_2/Al_2O_3$ higher than 3.1 and a crystal size of no less than 20 microns.

In conventional conditions, during hydrothermal preparation, NaY crystals with a size less than 1 micron, more often 0.1–0.3, are used. NaY crystals of this size are used for catalytic, sorption, and other areas of the oil refining and chemical industries [US3130007 patent (B1); Patent JP2001058816 (A), the IPC: S01V 39/24, publ. 06.03.2001; US6284218 (B1) Patent IPC: S01V 39/20, publ. 04.09.2001; Patent JP2008230886 (A), the IPC: B01J 29/08; S01V 39/02, publ. 02.10.2008].

Russian patent 2463108 discloses a method for producing aluminosilicate composition comprising from 40 to 75% by weight. Silicon oxide, the process of which comprises of (a) combining in a mixing zone water and aluminum sulfate to produce a mixture with a pH in the range from 1.5 to 6.5; (b) then, raising the pH of said mixture to a value in the range from 7.5 to 12 by adding sodium silicate to said mixture in said mixing zone, and (c) recovering a precipitate solid from said mixture in said mixing zone, wherein said precipitate solid comprises said aluminosilicate composition.

A method of producing a synthetic zeolite microsphere granule (A.S USSR №146285, 1968) includes spray drying an aqueous slurry of the crystalline zeolite to the binder addition (highly plastic clays and fine), taken in an amount of 10–35 wt.% of zeolite mass.

U.S. Patent No. 6872685 describes a composition of amorphous aluminosilicate, characterized surfactants bulk Si/Al ratio (SB ratio) in a range from 0.7 to 1.3 and containing less than 10% crystalline alumina phase. This amorphous aluminosilicate is obtained by mixing a silicate (sodium silicate) and an acid aluminum salt solution (aluminum sulfate) while maintaining the pH of the stirred solution less than 3, then by gradually adding a basic precipitant to the stirred solution to form a precipitated co-gel, which is then extracted, washed, and subjected to spray drying.

U.S. Patent No. 4988659 described aluminosilicate to gel and process for preparing such a gel. Co-gel was prepared by adding a silicate solution to an aqueous solution of the acid and acidic aluminum salt such as aluminum chloride or aluminum sulfate to form an acidified silica sol with a pH in the range from 1 to 4, further increasing the pH of the sol by adding a base, aging, and recovering the resulting sol to gel. The recovered co-gel can be further processed by using acid during excitation syneresis, followed by washing and then spray dried.

Despite a significant amount of research on methods obtaining synthetic zeolites, the issues of their application in the construction industry have not been fully covered. Meanwhile, the presence of amorphous aluminosilicate in the composition of synthetic zeolites creates prerequisites for their use in lime composites.

During the interaction of amorphous aluminosilicate with lime, the formation of calcium aluminosilicate and calcium hydrosilicates is possible, which will accelerate the curing and increase the strength of lime composites.

The purpose of the work is to develop technology for obtaining an additive based on synthetic aluminosilicates and evaluate the effectiveness of its use in dry lime mortars.

## 2. Materials and Methods

The following materials were used to prepare the supplement: technical aluminum sulfate, liquid sodium glass (GOST 13078-2021).

Solutions of sodium water glass and aluminum sulfate of various densities were prepared, after which the water glass solution was added to the aluminum sulfate solution.

The precipitate was thoroughly washed and filtered on White Tape filter paper. Then, it was placed in an oven and dried at a temperature of 100–110 °C to a constant weight, after which it was ground into powder.

To study the regularities of the influence of additives based on synthetic aluminosilicates on the properties of lime composites, samples were prepared on the fluffy lime of the 1st grade with an activity of 84.4%. For comparison, additives were used—diatomite, meta kaolinite. The content of additives based on synthetic aluminosilicates, diatomite, and metabolite was 3–30% by weight of lime. Compositions were prepared with a water–lime ratio W/L equal to W/L = 1.25. The samples hardened in air-dry conditions at a temperature of 18–20 °C and a relative air humidity of 60–70%.

Analysis of the granulometric composition of additive based on synthetic aluminosilicates was performed using an automatic laser diffractometer Fritsch Particle Sizer Analysette 22 (Fritsch, Bayern, Germany)

To determine the chemical composition, X-ray fluorescence analysis was performed using X-ray station ARL 9900 X-ray Work Station (Thermo Scientific, Waltham, MA, USA). X-ray diffraction spectra were obtained using $\lambda CoK\alpha1,2$ radiation. Diffraction angle interval $2\theta = 12–80\theta$, scanning step $0.02\theta$. For the unambiguous identification of the synthesis products, a full-profile method of quantitative X-ray phase analysis was used, which, along with the Rietveld algorithm, implements the Derivative Difference Minimization algorithm [10,11]. This algorithm is based on minimizing the local derivatives of the difference curve of the experimental and calculated diffraction spectra and eliminates the need for an approximation or numerical description of the background component of the X-ray diffraction pattern.

Structural data for model mineral compositions were used from the International Structural Database (ICSD). The concentration of the amorphous phase was determined using full-profile XRF with internal standardization. Anatase at a concentration of 30% by weight was used as a reference.

## 3. Results

We studied the influence of the mode of introduction and the amount of aluminum sulfate, the pH of the mixture, and the modulus of liquid glass on the yield of the finished product (Table 1).

**Table 1.** Aluminosilicates synthesis regimes.

| No. of Synthesis Mode p/p | pH of Aluminum Sulfate Solution $Al_2(SO_4)_3$ | Liquid Glass Module a * | Filtrate pH | Finished Product Yield, % |
|---|---|---|---|---|
| 1 | 5.0 | 2.69 | - | - |
| 2 | 3.0 | 2.69 | 9.00 | 41.90 |
| 3 | 1.50 | 2.69 | 9.00 | 38.90 |
| 4 | 5.00 | 2.88 | - | - |
| 5 | 3.00 | 2.88 | 9.00 | 34.00 |
| 6 | 1.50 | 2.88 | 9.00 | 42.90 |
| 7 | 5.00 | 2.69 | - | - |
| 8 | 3.00 | 2.69 | 5.00 | 40.48 |
| 9 | 1.50 | 2.69 | 5.00 | 44.76 |
| 10 | 5.00 | 2.88 | - | - |
| 11 | 3.00 | 2.88 | 5.00 | 33.12 |
| 12 | 1.50 | 2.88 | 5.00 | 45.87 |

Note: * Density of liquid glass was $\rho$ = 1279 kg/m$^3$.

It has been established that, at a pH level of aluminum sulfate solution $Al_2(SO_4)_3$ equal to pH $\geq$ 5, no precipitate is formed. The minimum yield of the product is 33.12% and is observed at the pH of the solution of aluminum sulfate $Al_2(SO_4)_3$ equal to pH = 3; moreover, the maximum yield (45.87%) is observed at the pH of the solution of aluminum sulfate $Al_2(SO_4)_3$ equal to pH = 1.5. An increase in the liquid glass modulus from 2.69 to 2.88 leads to an increase in the yield of finished products from 44.76% to 45.87%.

As a criterion for the optimal technology for obtaining the additive, the compressive strength of the lime composite was recorded, since the additive was subsequently intended for use in lime systems. Table 2 shows the results of the studies of the dependence of the strength of a lime composite with an additive obtained at different densities of liquid glass. Lime of the first grade with an activity of 84.4% was used, and the water–lime ratio is W/L = 1.25.

**Table 2.** Dependence of the strength of the lime composite with an additive synthesized at different densities of liquid glass.

| No. | Liquid Glass Density, kg/m$^3$ | Compressive Strength of Lime Composite, MPa |
|---|---|---|
| 1 | 1368 | 3.25 |
| 2 | 1279 | 2.61 |
| 3 | 1180 | 1.82 |
| 4 | 1113 | 1.76 |
| 5 | 1038 | 1.61 |

Note: The data were obtained at the pH of aluminum sulfate $Al_2(SO_4)_3$ solution equal to pH = 3.

It has been established that the maximum strength of lime specimens with an additive based on synthetic aluminosilicates, which is R = 3.25 MPa, is observed at a density of liquid glass used in the synthesis of the additive, equal to $\rho$ = 1368 kg/m$^3$.

The German standard DIN 18550 (part 2) notes that durability and resistance to external influences, as well as high crack resistance, are ensured when the plaster mortar has a compressive strength in the range of 2 to 5 MPa. Solutions with such strength characteristics are able to adapt to small deformations and resist cracking. Based on this, for the sake of economy, the optimal mode was chosen: the density and modulus of liquid glass are, respectively, $\rho$ = 1279 kg/m$^3$, M = 2.88, pH of aluminum sulfate $Al_2(SO_4)_3$ solution pH = 3 [12,13].

The resulting product is a white powder with a specific surface area determined using the Brunauer–Emmett–Teller (BET) method, Ssp = 86.5 m$^2$/g. The chemical composition of additives based on synthetic zeolite is presented in Table 3.

Analyzing the data obtained in Table 3, a high content of the chemical elements O, Si, and Na was revealed, amounting to 46.47–61.58%, 20.78–39.60%, and 5.54–16.52%, respectively, which indicates the predominance of oxides of the corresponding elements. The oxide composition is presented in Tables 3 and 4. It has been established that silica oxides predominate, constituting 56.21–55.45% (Table 4).

**Table 3.** The content of chemical elements in the additive based on synthetic aluminosilicates, %.

| The Content of Chemical Elements in Weight% | O | Na | Al | Si | S |
|---|---|---|---|---|---|
| Maximum | 62.58 | 16.52 | 8.05 | 39.60 | 8.94 |
| Minimum | 46.67 | 5.54 | 1.88 | 20.78 | 0.89 |

**Table 4.** The content of oxides presents in the additive based on synthetic aluminosilicates.

| Name Oxide | Content, % |
|---|---|
| $SiO_2$ | 56.21–55.45 |
| $Al_2O_3$ | 21.24–7.58 |
| $SO_3$ | 16.79–8.91 |
| $Na_2O$ | 19.10–13.91 |
| CaO | 0.15–0.0938 |

The X-ray diffraction pattern of the material is characterized by the presence of selective diffraction reflections and a pronounced structured background ("diffuse halo"). Based on X-ray diagnostics performed using the PDF 2 database of the International Center for Diffraction Data (ICDD) using the SearchMatchv.2 program, it was found that the crystalline part of the additive based on synthetic zeolite is represented by thenardite, a rhombic modification of sodium sulfate $Na_2SO_4$ (PDF 70-1541). In addition, a weak reflection at $2\theta = 21.29°$ was attributed to gibbsite (PDF 70-2038) (Figure 1).

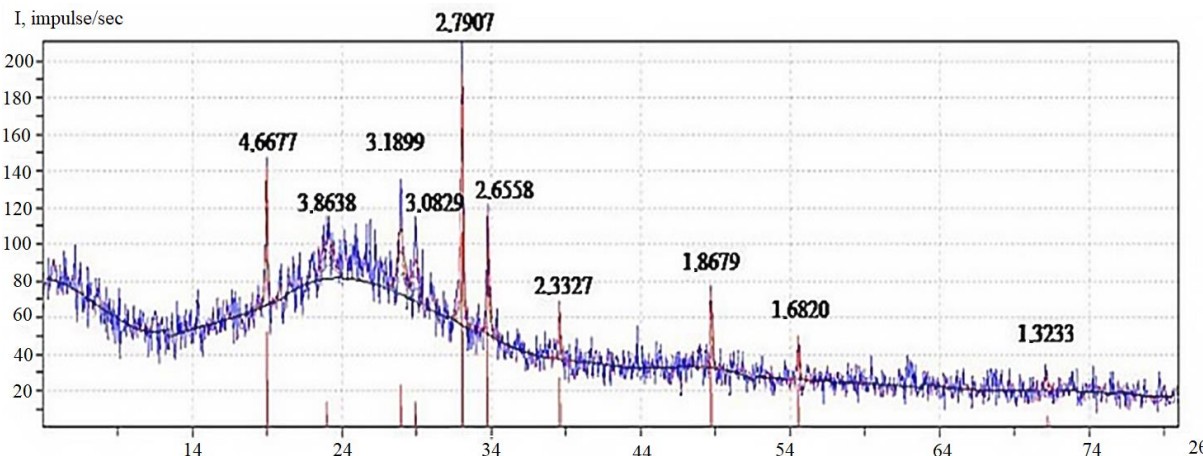

**Figure 1.** X-ray pattern of additives based on synthetic aluminosilicates.

X-ray diffraction analysis (XRF) revealed that the composition of the additive is represented by the following minerals: thenardite $Na_2SO_4$ (d = 4.6677; d = 3.8638; d = 3.0829; d = 2.3327); gibbsite $Al(OH)_3$ (d = 3.1899; d = 1.6820); minerals of the zeolite group (d = 2.7907; d = 2.6558; d = 1.8679) (Figure 1) [14–16].

Figure 2 shows the result of quantitative XRF in the variant of calculating the concentrations of crystalline phases. The results obtained are consistent with the chemical analysis data:

$$C_{amorph} = \sum C_i - (C_{Na2O} + C_{SO3}) = 77.5\% \tag{1}$$

The results obtained are in good agreement with the data of the authors [17,18].

The question of the structural nature of the resulting amorphous phase of the aluminosilicate composition requires separate consideration. According to [19], the characteristic that determines the structural motif of silica in amorphous phases can be the Si/O ratio, which is defined as the degree of silica connectivity in silicate clusters of the glass phase.

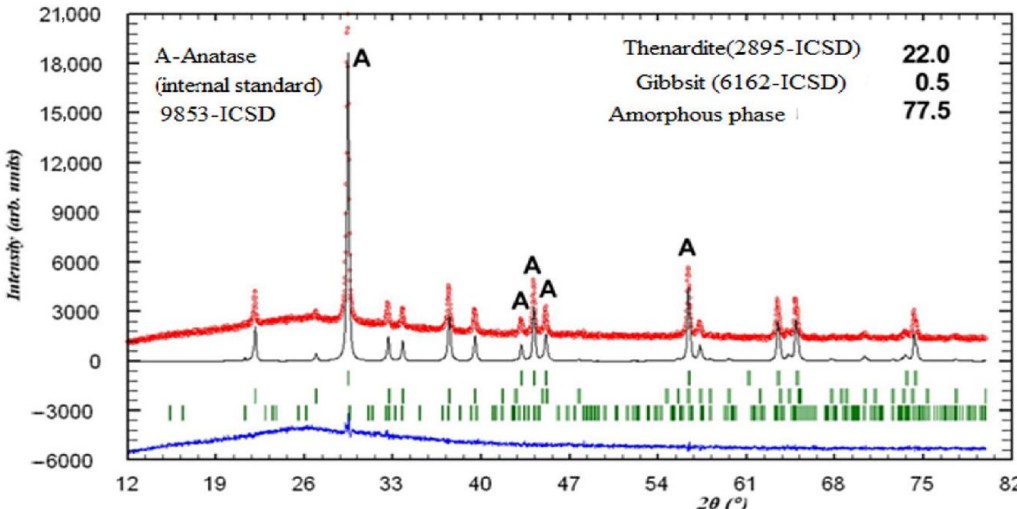

| | |
|---|---|
| Thenardite(2895-ICSD) | **22.0** |
| Gibbsit (6162-ICSD) | **0.5** |
| Amorphous phase | **77.5** |

**Figure 2.** The result of the full quantitative XRF.

The coefficient $f_{Si}$ is equal to the ratio of the number of silicon ions to the number of oxygen ions and expresses the degree of bonding of the silicon–oxygen framework. The coefficient $f_{Si}$ reflects the nature of the structure of silicates and the state of structural units, namely, silicon–oxygen tetrahedra. The type of silicon–oxygen radical according to the Si/O ratio is continuous in three dimensions; in this case, $f_{Si}$ is equal to 0.5. This indicates the framework structure of silica clusters in the amorphous material.

The microstructure of the resulting additive was studied using an electron microscope at a magnification of 20.000 times (Figure 3).

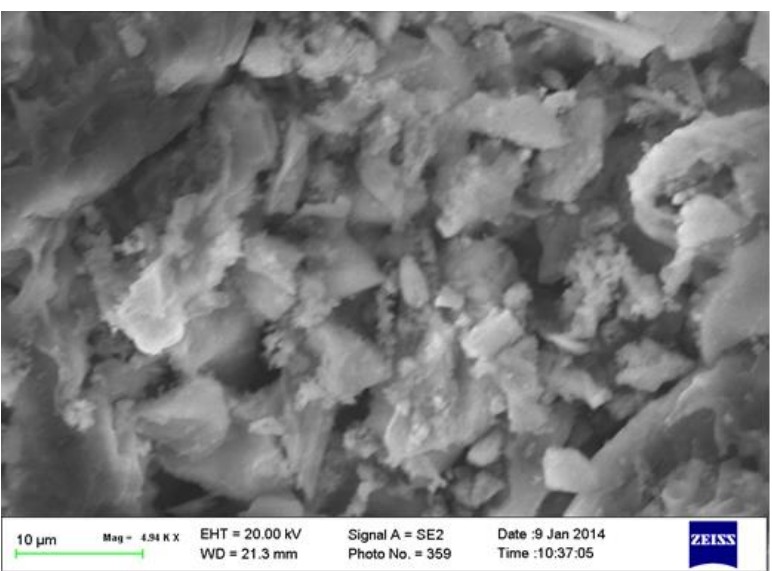

**Figure 3.** Microstructure of additives based on synthetic aluminosilicates ×20,000.

It has been established that the structure of the additive is represented mainly by particles, the size of which is 2.25–8.1 nm.

Table 5 shows the activity values of additives based on synthetic aluminosilicates, depending on the mode of its production.

**Table 5.** Properties of additive based on synthetic aluminosilicates.

| Synthesis Mode | Mixture pH | Suspension pH Additives | Solubility M, % | Activity A, mg/g |
| --- | --- | --- | --- | --- |
| 1 | 4.83 | 10.01 | 89.03 | 350 |
| 2 | 6.31 | 10.24 | 74.36 | 350 |
| 3 | 9.26 | 10.48 | 74.16 | 350 |
| 4 | 10.50 | 10.50 | 67.63 | 350 |

Additives based on synthetic aluminosilicates are characterized by the high activity of more than 350 mg/g. The solubility of the additive prepared according to the first mode is M = 89.03%, the additive prepared according to the fourth mode is M = 67.63%, and the cation exchange capacity is E = 38.7–43.51 mg·eq. An analysis of particle size distribution, performed using an automatic laser diffractometer Fritsch Particle Sizer Analysette 22, showed that less than 0.01% are particles with a size of 0.010–0.500 μm, and that the content of particles with a size of 100.000–200.000 μm is 0.44%. Less than 5% are particles with a diameter of 3.226 microns; less than 15% are particles with a diameter of 6.985 microns (Table 6).

**Table 6.** Granulometric composition of the additive.

| Fraction, μm | The Percentage Content, % |
| --- | --- |
| 0.01–0.50 | 0.01 |
| 0.50–2.00 | 1.81 |
| 2.00–3.00 | 2.55 |
| 3.00–4.00 | 2.8 |
| 4.00–5.00 | 2.73 |
| 5.00–10.00 | 12.61 |
| 10.00–20.00 | 16.61 |
| 20.00–45.00 | 27.2 |
| 45.00–80.00 | 29.14 |
| 80.00–100.00 | 4.09 |
| 100.00–200.00 | 0.44 |

To assess the sorption properties of the additive, sorption moisture was determined. When studying the kinetics of moisture sorption, the samples were predried in an oven at a temperature of t = 110 °C to a constant weight and placed in desiccators with different relative humidity φ = 18–97% and a constant temperature t = 20 ± 2 °C. Based on the data obtained, sorption and desorption isotherms were constructed (Figure 4).

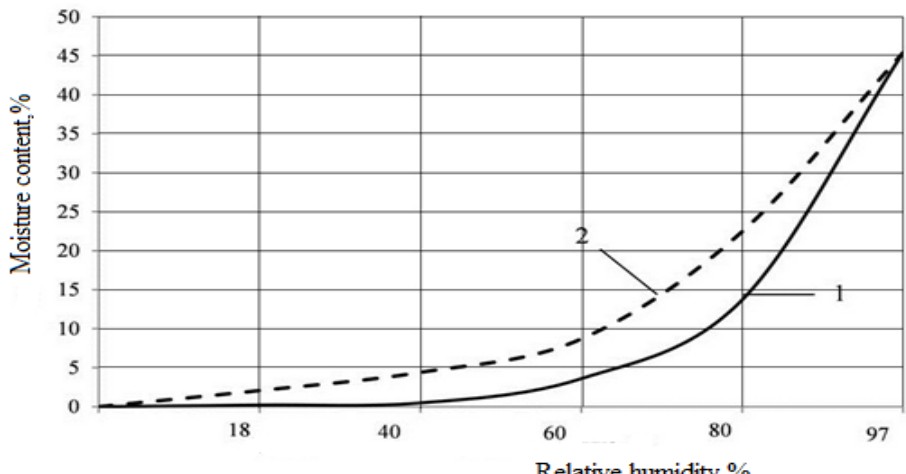

**Figure 4.** Sorption and desorption isotherms of an additive based on synthetic aluminosilicates: 1—sorption isotherm; 2—desorption isotherm.

Moisture sorption in the range of relative air humidity up to 18% obeys Henry's law, i.e., the dependence of sorption humidification on the value of relative air humidity is

close to linear. With an increase in the relative humidity of the air up to 40%, the moisture content in the samples increases in accordance with the Freundlich equation. The convex part of the isotherms ($\varphi$ = 60–80%) indicates the presence inside the studied samples of only adsorbed moisture, consisting of a single layer of water vapor molecules. An increase in the relative air humidity to 90% leads to the formation of adsorbed moisture films on the inner surface of the material, consisting of many layers of molecules. Starting from 80% humidity, there is a sharp increase in sorption moisture, which indicates the occurrence of the process of capillary condensation.

Sorption and desorption isotherms coincide only at very low and very high values of relative air humidity; at other values, they do not coincide. The sorption isotherms are located lower than the desorption isotherms, and the equilibrium moisture content at the same value of the relative air humidity during moisture desorption is less than during moisture sorption. The bulk density of the additive based on synthetic aluminosilicates is $\rho_{set}$ = 231 kg/m$^3$, and the true $\rho_{act}$ = 568.515 kg/m$^3$.

An analysis of the experimental data indicates that, with the introduction of an additive based on synthetic aluminosilicates in an amount of 10%, the strength is 94.25% higher than that of the control sample; with the introduction of meta kaolinite in an amount of 10%, the strength increases by 26.35% compared with the control sample; moreover, with the introduction of diatomite in an amount of 10%, the strength compared with control samples decreases by 6.08% (Table 7).

**Table 7.** The value of the compressive strength of lime composites with additives after 28 days.

| Additive | Percentage, % | | | | |
| --- | --- | --- | --- | --- | --- |
| | 3 | 5 | 10 | 20 | 30 |
| | Compressive Strength, MPa | | | | |
| Control (lime:water) | | | 1.48 | | |
| Diatomite | 0.952 | 1.23 | 1.39 | 2.46 | 2.76 |
| Meta kaolinite | 2.325 | 2.45 | 3.35 | 3.85 | 4.5 |
| Additive based on synthetic aluminosilicates | 2.275 | 2.48 | 2.875 | 3.1 | 3.54 |

However, the preparation of meta kaolinite (roasting at a temperature of 900 °C) is an energy intensive process; therefore, it is more appropriate to use an additive based on synthetic aluminosilicates.

X-ray diffraction analysis (XRD) revealed that, when an additive based on synthetic aluminosilicates is added to lime, the following are formed: zeolites (d = 3.036; d = 2.627; d = 1.687); portlandite Ca(OH)$_2$ (d = 4.911; d = 3.11; d = 1.876); calcite CaCO$_3$ (d = 2.491; d = 2.094; d = 2.491); calcium hydro silicate - sodium CaNaHSiO$_4$ (d = 1.926; d = 1.91; d = 2.287) (Figure 5) [20–27].

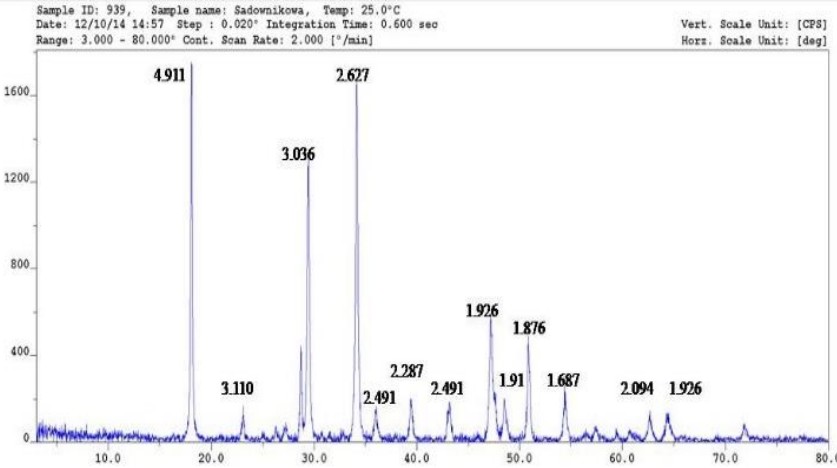

**Figure 5.** X-ray diffraction pattern of a lime composite in the presence of a synthetic zeolite additive.

The thermodynamic analysis of possible solid-phase isobaric–isothermal reactions was carried out in accordance with the second law of thermodynamics, which establishes a relationship between the thermal effect of a chemically irreversible process and the work of the corresponding irreversible process and is determined using the Gibbs–Helmholtz equation:

$$\Delta G = \Delta H_p + T \cdot \frac{\delta \Delta G}{\delta T} \tag{2}$$

where,

$\Delta G$ is the Gibbs energy;
$\Delta H_p$ is process enthalpy;
$T$ is temperature, K.

Considering that the synthesis of the additive proceeds at a temperature (293 K) close to the standard one (298 K), the thermal effect of the reaction and the change in the Gibbs energy were determined only at the standard state and were calculated as the difference between the sums of the corresponding indicators of the reaction products and starting substances. The results of the calculations of thermodynamic parameters are presented in Table 8. The results of the calculations indicate the probability of reactions proceeding in the forward direction. Large numerical values $\Delta G_{298}^0$ allow us to speak with sufficient probability about the possibility of these reactions occurring not only at the standard temperature (25 °C), but also at other temperatures.

**Table 8.** Thermodynamic parameters of isobaric–isothermal reactions for the production of synthetic aluminosilicates.

| Formula of the Compound | Heat of Formation, $\Delta H_{298}^0$ kJ/mol | Gibbs Energy of Formation, $\Delta G_{298}^0$ kJ/mol | Reactions |
|---|---|---|---|
| $Na_2SO_4$ | −1389.5 | −1271.7 | $Al_2(SO_4)_3 + 4Na_2SiO_3 + 2H_2O = 2NaAlSi_2O_6 \cdot H_2O + 3Na_2SO_4$ |
| $Al(OH)_3$ | −1275.7 | −1139.7 | $Al_2(SO_4)_3 + 3Na_2SiO_3 + 3H_2O = 2Al(OH)_3 + 3Na_2SO_4 + 3SiO_2$ |
| $NaAlSi_2O_6 \cdot H_2O$ | −682.68 | −732.48 | $Al_2(SO_4)_3 + 4Na_2SiO_3 + 2H_2O = 2NaAlSi_2O_6 \cdot H_2O + 3Na_2SO_4$ |
| $Na_2Al_2Si_3O_{10} \cdot 2H_2O$ | −1128.3 | −984.1 | $Al_2(SO_4)_3 + 4Na_2SiO_3 + 2H_2O = Na_2Al_2Si_3O_{10} \cdot 2H_2O + 3Na_2SO_4 + SiO_2$ |
| $NaAlSi_5O_{12} \cdot 3H_2O$ | −937.59 | −765.56 | $1.5Al_2(SO_4)_3 + 5Na_2SiO_3 + 6H_2O = NaAlSi_5O_{12} \cdot 3H_2O + 4.5Na_2SO_4 + 2Al(OH)_3$ |

The results of the thermodynamic calculations suggest that the most probable mechanism causing the hardening process is the formation of calcite, calcium sodium hydro silicate, minerals of the zeolite group, and portlandite.

Additional confirmation of the chemical interaction of the additive with lime is provided by the amount of chemically bound lime. It has been established that the amount of chemically bound lime in control samples (lime + water) at the age of 28 days of air-dry hardening is 46.5%, and, with the use of an additive based on synthetic aluminosilicates, it is 55.28% (Figure 6).

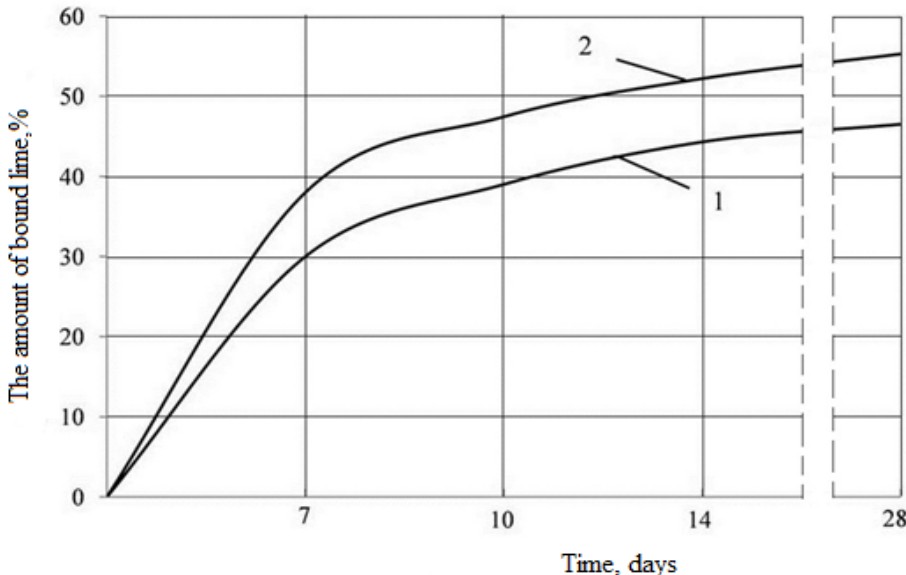

**Figure 6.** Change in the amount of bound lime during the hardening process 1—control; 2—with the use of an additive based on synthetic aluminosilicates.

### 4. Conclusions

A technology has been developed for obtaining an additive based on synthetic zeolites, which consists of the precipitation of aluminosilicates from sodium water glass with aluminum sulfate $Al_2(SO_4)_3$. The optimal parameters of the technology for obtaining the additive were established: the density and modulus of liquid glass are, respectively, $\rho = 1279$ kg/m$^3$ and M = 2.88, the pH of aluminum sulfate $Al_2(SO_4)_3$ pH = 3.

X-ray diffraction analysis and thermodynamic calculations have established that the mineralogical composition of the crystalline phase of the additive based on synthetic aluminosilicates is represented by the thenardite, gibbsite, and minerals of the zeolite group. It has been shown that the content of the amorphous phase is 77.5%. It was revealed that the additive based on synthetic aluminosilicates is characterized by high activity, which is more than 350 mg/g. An analysis of the granulometric composition, performed using an automatic laser diffractometer Fritsch Particle Sizer Analysette 22, revealed that less than 0.01% are particles with a size of 0.010–0.500 μm, and that the content of particles with a size of 100.000–200.000 μm is 0.44%. Less than 5% are particles with a diameter of 3.226 microns, and less than 15% are particles with a diameter of 6.985 microns. The true density of the synthesized aluminosilicates is $\rho$est = 568.515 kg/m$^3$, and the bulk density $\rho_{sat} = 231$ kg/m$^3$; moreover, the cation exchange capacity is E = 38.7–43.51 mg eq.

It was found that the introduction of an additive based on synthetic aluminosilicates into the formulation of a lime dry mixture accelerates the curing of coatings. The optimal concentration of the additive was selected, which is 10% by weight of lime. It has been shown that the introduction of an additive based on synthetic aluminosilicates into the lime–sand composition contributes to an increase in compressive strength at the age of 28 days of air-dry hardening by 1.9 times.

X-ray diffraction analysis, thermodynamic calculations, and chemical analysis revealed that samples based on lime compositions with an additive based on synthetic aluminosilicates are characterized by the content of the calcium–sodium hydrosilicates, calcite, calcium hydroxide, and minerals of the zeolite group, an increase in the amount of chemically bound lime by 8.74%.

**Author Contributions:** Conceptualization, V.I.L.; methodology, V.I.L.; software, V.I.L.; validation, V.I.L., K.V.Z. and M.A.S.; formal analysis, K.V.Z.; investigation, M.A.S.; resources, K.V.Z.; data curation, M.A.S.; writing—original draft preparation, M.A.S.; writing—review and editing, K.V.Z.; visualization, V.I.L., K.V.Z. and M.A.S.; supervision, K.V.Z.; project administration, V.I.L. All authors have read and agreed to the published version of the manuscript.

**Funding:** This research received no external funding.

**Data Availability Statement:** The data that support the findings of this study are available on request from the corresponding author.

**Conflicts of Interest:** The authors declare no conflict of interest.

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
