# Peer review of "Additive Based on Synthetic Aluminosilicates for Dry Lime Construction Mixtures"

_2673-7167, doi:10.3390/physchem3010013_

Round 1
Reviewer 1 Report
Thank you for the manuscript. Currently manuscript is difficult to understand from the perspective of implementation in the construction industry. Please refer to the following comments.
1. The title needs to be revised to make it more relevant.
2. The introduction should be enhanced with more relevant studies and improved for clarity.
3. Please correct typos and grammatical. Also, use consistent decimal places.
4. Please clearly specify the control and its values.
5. The quality of the XRD spectrograph is very low and it does not reflect the information related to lines 147-153.
6. Lines 178-180: Write clearly.
7. Currently, it is difficult to understand several aspects of the manuscript e.g., plenty of information is added within 132-133.
8. Methods are not clear, method description should be added in the method section, not in the result.
9. How was the elemental composition measured?
10. Please add SEM results as these will be very useful, if possible.
11. Please refer to the particular thermodynamic calculations (lines 228-230)
12. Can few findings be corroborated through literature such as 77.6% amorphous phase?
13. Reference 16, 21. Wouldn’t et al. be more appropriate for the remaining authors?
Author Response
1-8, 10, 12, 13 - we corrected
9. In section 2. "Materials and Methods" indicated the methodology. To determine the chemical composition, X-ray fluorescence analysis was carried out for X-ray station ARL 9900 X-ray Work Station (Thermo Scientific)
11. Made thermodynamic calculations
Reviewer 2 Report
This paper presents the an additive dry construction mixtures based on synthetic zeolites into the formulation of a lime mixtures to improve the compressive strength. There are a few clarifications the authors need to make and add additional information to the paper before accepting it for publication which include:
1. In the title mentioned mixture based on synthetic Aluminosilicates, but in the abstract the mixture is based on synthetic zeolites. Please consider changing the title or the abstract based on which mixture used in the study to make it clearer.
2. The authors should distinguish the contribution of this work from that of existing similar works. Add or expand the last paragraph of the introduction to clearly state what is new in this paper and contribution to practice in this field.
3. Section 2. Materials and Methods should be rewritten, before jumping to the result section.
4. The authors provide detailed results from their experimental studies through several additive mix-percentages. However, there is no control mix-design without a mixture to compare the results presented. Providing this information will make the study reasonably acceptable.
Author Response
1-3 we corrected
4. Figure 6. Change in the amount of bound lime during the hardening process 1 - control; 2 - with the use of an additive based on synthetic aluminosilicates, Table 9.
Reviewer 3 Report
Authors performed research focused on production of synthetic zeolites as additives for dry construction mixtures.
Very simple introduction, supported by a few references. Aims clearly stated.
Methodological approach is very poor, giving just little information, no supporting references and/or Norms and/or Protocols at all!
Results presentation is, in general, clear.
No true discussion is done; confrontation with results obtained by others must be done.
Conclusions are coherent with obtained results but can be strengthened after improving discussion as required.
Author Response
all comments corrected
Round 2
Reviewer 1 Report
- Line 205: Is it 77,5 or 77.5%?
- In view of above comment, overall, please use consistent SI notations for numbers
- SEM discussion can be further enhanced.
- Remove typos
- Please be sure that referencing should be according to journal guidelines.
Author Response
We corrected
Reviewer 2 Report
Thank you for revising the manuscript. It looks good now.
Author Response
We corrected
Reviewer 3 Report
Authors revised paper taking into account major reviewer comments/suggestions/questions.
Author Response
We corrected